# The Alterations in CD14 Expression in Periodontitis: A Systematic Review

Vivian Hirsch [1], Alice Blufstein [1,2], Christian Behm [1,3] and Oleh Andrukhov [1,*]

1 Competence Center for Periodontal Research, University Clinic of Dentistry, Medical University of Vienna, 1090 Vienna, Austria; vivian.hirsch@mail.mcgill.ca (V.H.); alice.blufstein@meduniwien.ac.at (A.B.); christian.behm@meduniwien.ac.at (C.B.)
2 Division of Conservative Dentistry and Periodontology, University Clinic of Dentistry, Medical University of Vienna, 1090 Vienna, Austria
3 Division of Orthodontics, University Clinic of Dentistry, Medical University of Vienna, 1090 Vienna, Austria
* Correspondence: oleh.andrukhov@meduniwien.ac.at

**Abstract:** Objective: Cluster of differentiation (CD14) is an important protein involved in activating toll-like receptors by bacterial components. It exists as either a transmembrane or soluble protein, called mCD14 or sCD14, respectively. Several studies show that CD14 regulates the inflammatory response to periodontal pathogens, and its expression is altered in periodontitis, an inflammatory disease of tooth-supporting tissues. It is the intent of this review to investigate the levels of expression of mCD14 and sCD14 in peripheral blood monocytes, saliva, gingival crevicular fluid, and gingival tissue biopsies in periodontitis patients. Methods: PubMed, Scopus, Ovid/Medline, Embase, and the Cochrane Library were consulted for the online literature search. To ensure methodical quality, titles and abstracts were reviewed in accordance to the PRISMA guidelines. Data extraction and evaluation of the full texts were executed in agreement with the GRADE approach. Results: This systematic review shows that mCD14 levels are decreased in peripheral blood monocytes of periodontitis patients in comparison to healthy patients, while sCD14 levels in sera, gingival crevicular fluid (GCF), and biopsies of periodontitis patients have a tendency to be increased in comparison to healthy controls. The evaluation of CD14 in gingival biopsies and periodontal tissues elucidated the fact that interpretation of the data obtained with qPCR, ELISA, and flow cytometry is questionable.

**Keywords:** CD14; mCD14; sCD14; inflammation; periodontitis





## 1. Introduction

Periodontitis is a multifactorial, inflammatory disease that causes the destruction of the periodontal tissues and, as the ultimate endpoint, can even lead to the loss of the tooth [1]. According to a recent study by Tonetti, et al. [2], the global burden of periodontal diseases remains high, with the most severe form affecting 11.2% of the world's population [3]. *Porphyromonas gingivalis,* a gram-negative bacterium, is currently considered a keystone pathogen [4] and shows a strong association with periodontitis [5]. Recent studies have underscored that the pathogenicity of *P. gingivalis* lies in its constituents: Lipopolysaccharide (LPS), gingipains, and fimbriae/pili [6]. Each factor has been widely studied, and integral roles in the progression of periodontitis have been elucidated [6]. The human innate immune system comprises a family of pattern recognition receptors, termed toll-like receptors (TLRSs), which recognize various bacterial components and initiate the inflammatory reaction in response [7]. The recognition of bacterial components by TLRs is facilitated by numerous co-receptors, with the cluster of differentiation (CD) 14 being one of them [8].

Originally described as a myeloid differentiation antigen detected on mature monocytes/macrophages [9] and neutrophils, the CD14 molecule is a key factor for the innate recognition of bacteria [9,10]. CD14, a 55-kDa glycoprotein, exists in two forms, a soluble

form (sCD14) [11] and a membrane bound form (mCD14) [11]. The membrane bound form is found mainly on mature monocytes, macrophages, and activated neutrophils, and is immobile due to the glycosylphosphatidylinositol tail [10,11]. The circulating form found in serum and other body fluids, is thought to be the result of protease-mediated shedding/cleavage of mCD14 [12]. sCD14 provides cells such as fibroblasts, and epithelial and endothelial cells that lack the membrane bound form of CD14, a sensitivity to bacterial products [12]. Thus, ultimately both mCD14 and sCD14 can function as a receptor for LPS of gram-negative bacteria and for various cell wall products of gram-positive bacteria [10]. In addition to its role in LPS/cell wall products signaling, sCD14 might play a part in inflammatory diseases by regulating the immune systems response [13]. In periodontal derived mesenchymal stromal cells, sCD14 drastically enhances inflammatory response to various bacterial components, such as lipopolysaccharide, lipoteichoic acid, and lipoproteins [14,15].

Since CD14 is an important regulator of the inflammatory response to bacteria, it is hypothesized to play an essential role in regulating periodontitis pathogenesis. However, the exact role of CD14 in periodontal health and periodontitis is not well defined. This systematic review aims to examine the currently available literature with reference to CD14 and periodontitis, focusing on how its role pertains to the pathogenesis of periodontitis. This review explicitly explores the levels of expression of mCD14 and sCD14 in peripheral blood monocytes, saliva, gingival crevicular fluid, and gingival tissue biopsies to allow a more in-depth understanding of the mechanism of periodontitis and provide information for future outcome measures for periodontal disease.

## 2. Materials and Methods

### 2.1. Eligibility Criteria

Experimental investigations, randomized controlled clinical trials, and available literature were screened with reference to CD14 and periodontal disease. In accordance with the Cochrane standards for systematic review, the Preferred Reporting Items for systematic reviews and Meta-Analysis (PRISMA) guidelines were adhered to [16].

### 2.2. Search Strategy for Identification of Studies

To identify preliminary relevance for studies to be included, a comprehensive research methodology was developed for each database. The search terms (MeSH terms) were "CD14", "sCD14"," soluble CD14"," mCD14", "cluster of differentiation 14" AND "periodontal disease", "periodontitis", "periodontal pathogenesis" to find as many relevant articles as possible.

### 2.3. Electronic Literature Search

Five databases: PubMed, Ovid/Medline, Embase, The Cochrane Library, and Scopus; were searched. The search was conducted from October 2019 to December 2020. All studies published until December 2020 were considered. No language restriction was set, however, included studies were solely written in English. The titles and records were screened for duplicates and consequently assessed in regard to the specified inclusion and exclusion criteria, as well as relevancy to topic. After fulfilling these benchmarks, full texts were examined and re-assessed. Furthermore, by manually reviewing lists of references of other scientific articles, supplementary records were sought for. Grey literature was not included.

Screening: Inclusion and Exclusion Criteria

The full texts were included if they pertained to periodontal disease and additionally explored the relation with CD14. The eligibility criteria were liberally applied at the start to warrant as many studies as possible, and no study was excluded without assessment. In developing a pool of studies that is relevant to the purpose of this review, these inclusion and exclusion criteria were specifically set forth. The study population should be human samples specifically (gingival crevicular fluid (GCF), Tissue, Serum, Blood), no sole in vitro

studies, cell culture, or animal studies. There were no limits set on gender, grade, language, region, or age of the study population. However, studies with diseased individuals (i.e., HIV, immune deficiencies, diabetic populations, etc.) that had no healthy controls were emitted. In regard to the nature of the intervention/reference test, no limits were set, though Western blots, flow cytometry analysis, enzyme-linked immunosorbent assay (ELISA), and polymerase chain reaction (PCR) were the reference tests of choice. Additionally, singular polymorphism PCR studies were excluded, as this was not the sole focus of the review and would overextend the scope of this paper.

### 2.4. Quality and Eligibility Assessment

To limit the risk of bias caused by insufficiencies in design, conduct, or analysis of the study, the studies were individually assessed in accordance to the Cochrane Reviewers Handbook. For this review, each record was assessed on four different points in accordance with the GRADE (Grades of Recommendation Assessment Development and Evaluation) approach [17]. This method describes a structured procedure that permits the appraisal of quality of evidence and provides a scale for the strength of recommendation, given a set of outlined factors [17]. In this review, we had a few moderate-quality studies, which are categorized as downgraded randomized trials, or upgraded observational studies, as well as studies that include minimal factors which could reduce the general findings. Multiple studies contained in this review were low quality studies, which through this approach are seen as double-downgraded randomized trails that contain numerous factors that weaken the overall judgement quality, as well as studies that are of similar quality, contradictory, or sole observational studies. Very Low GRADE quality of evidence is given to downgraded observational studies, case series, or case reports that lack of evidence. Additionally, it is strongly suggested that the quality level of evidence can be reduced by additional factors, which are lack of evidence, limitations in the design and precision, inconsistency, and risk of bias [17].

### 2.5. Data Synthesis and Sensitivity Analysis

In adherence to the above outlined procedure, 20 relevant studies that met the criteria were included: [18–37]. Detailed information of these studies is listed in Tables 1–4. The following predefined information points were obtained and summarized for each of the included studies:

- Study Details (Title, Author, Year)
- Study Design (Methods/Materials)
- Sample (Size and Type)
- Outcome and General Findings

Due to the heterogenicity of the findings a statistical analysis and/or additional pooling of the retrieved data was deemed infeasible. This difficulty in comparable parameters also made sensitivity analyses not possible. Therefore, only a narrative analysis of the compiled data seemed reasonable. The results were tabularized and classified into four sub categories (CD14 in peripheral monocytes, serum sCD14, saliva/GCF, and biopsy studies) and analyzed accordingly (Tables 1–4).

**Table 1.** Summary of studies on CD14 expression in peripheral blood monocytes.

| Study Details | Study Design | Samples | Findings | Grade |
|---|---|---|---|---|
| **Cheng et al. (2018) Comparative analysis of immune cell subsets in peripheral blood from patients with periodontal disease and healthy controls** | Peripheral venous blood was collected and various immune cell subsets including CD14$^+$ monocytes were identified based on ex vivo staining. Activation markers CD16, CD40, CD54, CD86, and HLA-DR were measured in CD14$^+$ monocytes. | **15** CP patients; **15** AP patients; **13** healthy controls | No significant differences in the frequencies of CD14$^+$ monocytes or in the phenotype of CD14$^+$ monocytes in peripheral blood between CP, AP, and healthy controls were found. | Moderate |
| **Nicu et al. (2008) Expression of FcγRs and mCD14 on polymorphonuclear neutrophils and monocytes may determine periodontal infection** | Peripheral venous blood was collected by venipuncture. Citrated whole blood (40μl) was stained with the fluorochrome-conjugated antibodies. Levels of FcγRI, IIa, III, CR3, and mCD14 on PMNs and monocytes were measured by flow cytometry. Activation of PMNs and monocytes in response to stimulation with Aa and Pg was assessed by means of change in mCD14 expression in flow cytometry analysis. | **19** periodontitis patients; **18** healthy controls | Bacterial stimulation assay showed that periodontitis was associated with an enrichment of the FcγRIII+ monocytes ($p = 0.015$) with concomitant low mCD14 ($p = ·0.001$). | Moderate |
| **Pietruska et al. (2006) Evaluation of mCD14 expression on monocytes and the blood level of sCD14 in patients with generalized aggressive periodontitis** | The expression of mCD14 in peripheral blood monocytes was determined by flow cytometry. | **16** generalized AP patients; **13** healthy controls | No statistically significant difference in the expression of mCD14 on monocytes between generalized AP patients and control subjects. | Moderate |
| **Nagasawa et al. (2004) Expression of CD14, CD16, and CD45RA on monocytes from periodontitis patients** | Peripheral venous blood was collected, and the expression of CD14 on monocytes was determined using flow cytometry. | **33** AP patients: **22** female, **11** males; **55** CP patients: **35** females, **20** males; **30** healthy controls, **16** females, **14** males | The percentage of CD14$^{bright}$ monocytes in chronic periodontitis patients was significantly lower than healthy subjects (chronic periodontitis patients: 70 ± 2.5%; healthy subjects: 82 ± 1.7%, $p < 0.05$). The mean percentage of CD14$^{bright}$ monocytes in aggressive periodontitis patients (70 ± 2.5%) was also lower than healthy subjects, but the difference was not statistically significant. | Moderate |

Table 1. *Cont*.

| Study Details | Study Design | Samples | Findings | Grade |
|---|---|---|---|---|
| **Buduneli et al. (2001) Flow-cytometric analysis of lymphocyte subsets and mCD14 expression in patients with various periodontitis categories** | Three-color flow cytometry and a panel of relevant monoclonal antibodies were used to determine the percent expression of various cell surface markers on peripheral blood mononuclear cells (PBMCs). | **22** early onset periodontitis patients, age 15–35 years; **10** adult periodontitis patients, age 38–57 years; **13** systemically and periodontally healthy controls, age 22–54 years. | The level of mCD14 in early onset periodontitis patients (7.18%) was lower than that of adult periodontitis patients (9.3%) and the control subjects (9.2%), but the differences were not statistically significant. | Moderate |
| **Shapira et al. (1996) Prostaglandin E2 secretion, cell maturation, and CD14 expression by monocyte-derived macrophages from localized juvenile periodontitis patients** | Peripheral venous blood was collected from localized juvenile periodontitis patients and control subjects. mCD14 was determined by In-cell ELISA; CD71 was chosen as macrophage maturation antigen. | **8** localized juvenile periodontitis patients; **8** age and sex matched healthy controls | mCD14 expression on monocytes in periodontitis patients was decreased compared to controls. After 5 or 10 days in culture, monocytes from both control and LJP subjects expressed comparable amounts of CD14. | Low Reason: Low sample size. |

Table 2. Summary of studies on sCD14 levels in serum.

| | | | | |
|---|---|---|---|---|
| **Hayashi et al. (1999) Increased levels of soluble CD14 in sera of periodontitis patients** | A serum sample was obtained from the median cubital vein of each patient during the initial examination. The concentration of CD14 was measured by a sandwich ELISA using two monoclonal antibodies (MAb) against different epitopes of sCD14 (IBL, Hamburg, Germany). | **38** Periodontitis patients, including **20** adult periodontitis patients; age 42–65 years and **18** early onset periodontitis patients, age 16–41 years; **25** healthy controls, age 20–34 years. | The sCD14 concentration in the serum was significantly higher in patients with periodontitis than in the healthy controls (3.22 versus 2.65 mg/L, $p < 0.01$). This was also observed when patients were classified as either adult or early onset periodontitis ($p < 0.01$). No significant difference between periodontitis groups was found. No correlation between the levels of sCD14 and the clinical variations of the patients was demonstrated. | Moderate |

**Table 2.** *Cont.*

| | | | | |
|---|---|---|---|---|
| **Nicu et al. (2009)** **Soluble CD14 in periodontitis** | Serum sCD14 concentration was measured by sandwich ELISA using two monoclonal antibodies (MAbs) against different epitopes of sCD14. | **59** moderate periodontitis patients; **46** severe periodontitis patients **57** healthy controls. The study population was derived from two previous studies. Included were all subjects for whom the plasma levels of sCD14, serology, and CD14260 genotype could be determined. | Tendency of higher sCD14 levels in periodontitis patients compared to controls was observed ($p = 0.061$). In a frequency distribution analysis using as threshold the 75th percentile of the sCD14 levels of the control subjects (2.03 mg/L), a higher number of moderate and severe periodontitis patients having sCD14 values above this threshold was observed compared to healthy controls (34% and 50.0%, respectively, versus 25%; $p = 0.026$). Furthermore, in periodontitis patients (both moderate and severe periodontitis), the sCD14 levels were positively correlated with the severity of periodontal destruction. | Moderate |
| **Raunio et al. (2009)** **Serum sCD14, polymorphism of CD14$^{-260}$ and periodontal infection** | Serum sCD14 concentration was analyzed using ELISA and CD14$^{-260}$ genotype using polymerase chain reaction (PCR). | **56** CP patients; **28** healthy controls. | Serum sCD14 levels were significantly higher in subjects with periodontitis than in control subjects (4.9μg/mL vs. 3.8 μg/mL, $p < 0.001$). Serum sCD14 levels were associated significantly with the extent of advanced periodontal disease. | Moderate |
| **Pietruska et al. (2006)** **Evaluation of mCD14 expression on monocytes and the blood level of sCD14 in patients with generalized aggressive periodontitis** | Serum sCD14 level was examined with ELISA method. | **16** generalized aggressive periodontitis patients; **13** healthy controls. | Similar serum sCD14 levels in periodontitis patients and healthy subjects. | Moderate |

**Table 3.** Summary of studies on sCD14 levels in saliva and gingival crevicular fluid (GCF).

| Study Details | Study Design | Samples | Findings | Grade |
|---|---|---|---|---|
| **Jin & Darveau (2001)** **Soluble CD14 levels in gingival crevicular fluid of subjects with untreated adult periodontitis** | Gingival crevicular fluid was collected from untreated adult periodontitis patients, sCD14 levels were determined by ELISA and presented as total amount (ng/site) and concentration (μg/mL). Plaque index, bleeding index, probing depth, and clinical attachment level were determined in the frame of periodontal examination. | **15** periodontitis patients. | sCD14 was detected in all 15 subjects and in 59% (62/105) of the sampled sites. The mean total amount of sCD14 was $1.71 \pm 0.40$, range 0.03 to 5.41 ng/site; the concentration of sCD14 was $14.04 \pm 4.15$, range 0.16 to 51.74 μg/mL. No significant difference in clinical data was found between the sites with and without detectable levels of sCD14. On the basis of the individual profile, 15 subjects were divided into a high sCD14 group and a low sCD14 group. Comparatively, the low group showed greater mean PD and a higher percentage of sites with PD $\geq 5.0$ mm ($p < 0.05$). Additionally, sCD14 concentrations showed a negative correlation with pocket depth (r = $-0.636$, $p = 0.0174$). | Low Reason: Missing Control Groups |
| **Duncan et al. (2004)** **Loss of lipopolysaccharide receptor CD14 from the surface of human macrophage-like cells mediated by *Porphyromonas gingivalis* outer membrane vesicles** | Gingival crevicular fluid was collected and analyzed for the presence of sCD14 receptor and CD14-derived fragments. Western immunoblotting analysis of GCF samples from patients affected by moderate or advanced periodontitis. | **24** individuals subdivided in three categories: 1. 8 healthy, PD < 3 mm 2. 8 moderate periodontitis, PD = 4–6 mm 3. 8 advanced periodontitis, PD >7 mm. | Western immunoblotting analysis of GCF samples showed the traces of CD14 in 3 out of 8 GCF samples from healthy subjects, and 9 out of 16 samples from periodontitis patients. Samples from patients affected by moderate or advanced periodontitis contained higher amount of sCD14. | Low Reason: Sample Size |
| **Isaza-Guzman et al. (2008)** **Estimation of sCD14 levels in saliva obtained from patients with various periodontal conditions** | Unstimulated whole saliva samples were obtained from patients with chronic periodontitis, aggressive periodontitis, and healthy controls. The levels of sCD14 were measured in saliva samples with ELISA. The periodontal status of each subject was assessed based on probing depth, clinical attachment loss, and the extent of periodontal breakdown. | **34** CP patients; **19** AP patients; **17** healthy controls. | Salivary sCD14 were significantly greater in the CP (mean = 15.57 ng/mL; $p = 0.011$) and AP (mean = 12.94 ng/mL; $p = 0.008$) patients compared with healthy controls (mean = 6.57 ng/mL). No significant difference in salivary sCD14 levels between CP and AP patients was observed. A highly significant correlation (Spearman) between data from salivary sCD14 levels and the clinical measurements of mean clinical attachment loss (r = 0.331, $p = 0.005$), mean pocket depth (r = 0.383, $p = 0.001$), extent of periodontitis (Percentage of periodontal pockets $\geq 4$ mm deep and attachment loss $\geq 2$ mm, r = 0.387, $p = 0.001$). | Moderate |

**Table 3.** *Cont.*

| Study Details | Study Design | Samples | Findings | Grade |
|---|---|---|---|---|
| **Prakasam et al. (2014)** **Evaluation of salivary biomarker profiles following non-surgical management of chronic periodontitis** | Unstimulated whole saliva was collected from periodontally healthy individuals and patients with chronic periodontitis at diagnosis and at 1- and 6-weeks following scaling and root planning (SRP). Several biomarkers including sCD14 were measured by ELISA. Western blot assays were used to confirm the presence/absence of sCD14. | **20** CP patients before and 6 weeks post scaling root planning (SRP); **20** healthy controls. | Salivary sCD14 levels were significantly higher (1.81-fold, $p = 0.018$) in CP patients as compared with healthy controls. sCD14 in saliva increased marginally in the beginning, following SRP (1.9-fold), there was a decreasing trend at week 6 post-SRP (1.47-fold) as compared to baseline. | Moderate |

**Table 4.** Summary of studies of CD14 expression in periodontal tissue biopsies.

| Study Details | Study Design | Samples | Findings | Grade |
|---|---|---|---|---|
| **Nibali et al. (2017)** **Leukocyte receptor expression in chronic periodontitis** | Human gingival biopsy were collected from diseased and control sites and processed by flow cytometry for the determination of the expression of 16 leukocyte receptors, including CD14. | **37** periodontitis patients. | Expression CD14 was higher in diseased compared to control sites ($p < 0.001$). Sampled sites with less bleeding on probing exhibited higher expression CD14 ($p = 0.011$). | Low Reason: Missing control group |
| **Sumedha et al. (2017)** **Immunohistochemical localization of TLR2 and CD14 in gingival tissue of healthy individuals and patients with chronic periodontitis** | CD14 in gingival tissues in chronic periodontitis (CP) and healthy patients. The expression of TLR2 and CD14 with the histological grades of CP was evaluated. 30 Gingival specimens from CP patients and 10 from healthy individuals were examined. Tissues from both groups were immunostained with antibodies against TLR2 and CD14. | **30** CP Subjects; **10** healthy controls. | TLR2 and CD14 expression was greatest in the periodontitis group (severe grade), followed by moderate and mild grades. Positive correlation of TLR2 and CD14 expression levels with the severity grades of chronic periodontitis. | Moderate |

**Table 4.** *Cont.*

| Study Details | Study Design | Samples | Findings | Grade |
|---|---|---|---|---|
| **Li et al. (2014)** **Differential expression of Toll-like receptor 4 in healthy and diseased human gingiva** | Human gingival biopsies were collected from healthy gingival or chronic periodontitis tissue. Primary human gingival keratinocytes (HGKs) and gingival fibroblasts (HGFs) were isolated and cultured. Transcripts of TLR4, MD-2, CD14, and LBP, and their protein products, were examined using RT-PCR, immunoprecipitation, and immunoblotting. | **34** samples from healthy subjects (42.2 ± 5.8 years old; 19 women); **44** gingival biopsies from advanced CP patients (46.3 ± 9.1 years old; 23 women). HGKs and HGFs were isolated from 13 control samples; of those, RT-PCR, immunoprecipitation and immunoblotting were performed on cells from 9, 7, and 10 samples, respectively. | Three TLR4 splicing variants, two MD-2 splicing variants and one CD14 mRNA, were expressed by gingival keratinocytes and fibroblasts. Expression of TLR4, CD14, and MD-2 proteins was detected in keratinocytes and fibroblasts in vitro. No qualitative difference in CD14 expression between cells isolated from periodontitis patients and healthy subjects was detected. | Moderate |
| **Scheres et al. (2011)** **Periodontal ligament and gingival fibroblasts from periodontitis patients are more active in interaction with** *Porphyromonas gingivalis* | Primary periodontal ligament and gingival fibroblasts from periodontitis patients and healthy control subjects were challenged in vitro with viable *P. gingivalis*. Gene expression of CD14 was assessed by real-time PCR. | **14** periodontitis patients; **8** healthy controls. | Periodontal ligament fibroblasts from periodontitis patients had a higher mRNA expression of CD14, both before and after *P. gingivalis* challenge. In gingival fibroblasts, no difference in CD14 expression between periodontitis patients and healthy controls was observed. | Low Reason: Sample Size |
| **Ren et al. (2005)** **The Expression profile of Lipopolysaccharide Binding Protein, Membrane Bound CD14, and Toll-Like Receptors 2 and 4 in Chronic Periodontitis** | Collection of gingival biopsies from subjects with chronic periodontitis, including periodontal pocket tissues (PoTs) and clinically healthy gingival tissues (HT-Ps), and from periodontally healthy subjects as controls (HT-Cs). The expression of CD14 and lipopolysaccharide bidning protein (LBP)was detected by immunohistochemistry and reverse transcription-polymerase chain reaction (RT-PCR). | **43** CP patients (22 males and 21 females; 22 to 65 years, mean age of 47.9 ± 3.7 years); **15** clinically healthy controls (7 males and 8 females; mean age of 23.4 ± 3.6 years). | LBP and mCD14 were simultaneously detected in 91% of PoTs, 85% of HT-Ps, and 100% of HT-Cs. LBP and mCD14 mRNAs were simultaneously detected in 55% of PoTs, 55% of HT-Ps, and 75% of HT-Cs. The expression of LBP was confined to the gingival epithelium, whereas mCD14 was observed around the epithelium-connective tissue interface. | Moderate |

**Table 4.** *Cont.*

| Study Details | Study Design | Samples | Findings | Grade |
|---|---|---|---|---|
| **Jin et al. (2004)** **The in vivo expression of membrane-bound CD14 in periodontal health and disease** | Gingival biopsies were obtained and CD14 was analyzed by immunohistochemistry. | **24** CP patients; **22** periodontal pocket tissue samples; **7** healthy controls. | mCD14 was detected in 21 of 22 periodontal pocket tissues and all other categories of tissues. mCD14 positive cells were mainly confined to the gingival epithelium-connective tissue interface. The expression levels in periodontally healthy subjects were significantly higher than in the patients. Within the patients, clinically healthy tissues showed greater levels of mCD14 than periodontal pocket tissues and granulation tissues. mCD14 was commonly expressed in both healthy and diseased gingival tissues and was predominantly confined to the epithelium-connective tissue interface. | Low Reason: Sample size and Design |
| **Roberts et al. (1997)** **Profile of cytokine mRNA expression in chronic adult periodontitis** | The prevalence of mRNA for inflammatory cytokines, including CD14, was tested in diseased and healthy gingival tissue. Gingival mononuclear cells or whole gingival biopsies from periodontitis patients and controls were obtained, and the mRNA expression of inflammatory cytokines, including CD14, was evaluated by reverse-transcription polymerase chain-reaction (RT-PCR) procedures. Additionally, tissue sectioning and immunohistochemical staining in periodontal tissue was also performed. | **32** periodontitis patients; **5** healthy controls. | CD14 mRNA was present in both healthy and diseased tissue samples (100%). Tissue section histochemistry showed that CD14 was detected in very low numbers in healthy biopsies (average of 1.9 cells/mm$^2$, range 0.8–3.0) and more prevalent in diseased biopsies (average of 31 cells/mm$^2$, range 0.4–140). CD14$^+$ cells were localized primarily in the connective tissue in close association with the lymphoid aggregates. | Moderate |

## 3. Results

Figure 1 shows a flowchart of the searching process. A total of 20 studies were identified for the inclusion in the review. Originally, the initial search in the databases of Medline, Embase, PubMed, Cochrane, and Scopus resulted in 1878 citations. After duplicates were removed, a total of 570 records remained. Of these 570, 338 were discarded because after the assessment of the abstract, the studies did not fit into the stipulated criteria. For the remaining 232 citations, the full text versions were obtained and examined more precisely. Multiple studies focused solely on the polymorphism of CD14, and thus were excluded [38–47]. Additional studies were excluded in the analysis because the levels of CD14 were not specifically retrieved in human patients, but in human cell cultures obtained from a Research center [48,49] and in vitro dental mesenchymal stromal cells or cell line studies [39]. Furthermore, subsequent reports were not included due to the fact that the tested or the control group were known to be individuals with compromised health (i.e., renal transplant patients [50,51] or HIV [52]. Finally, overview articles [53] and literature reviews of markers in periodontitis [54] provided great general information, but lacked the specificity in providing case control study examples, and thus results as such were excluded as well.

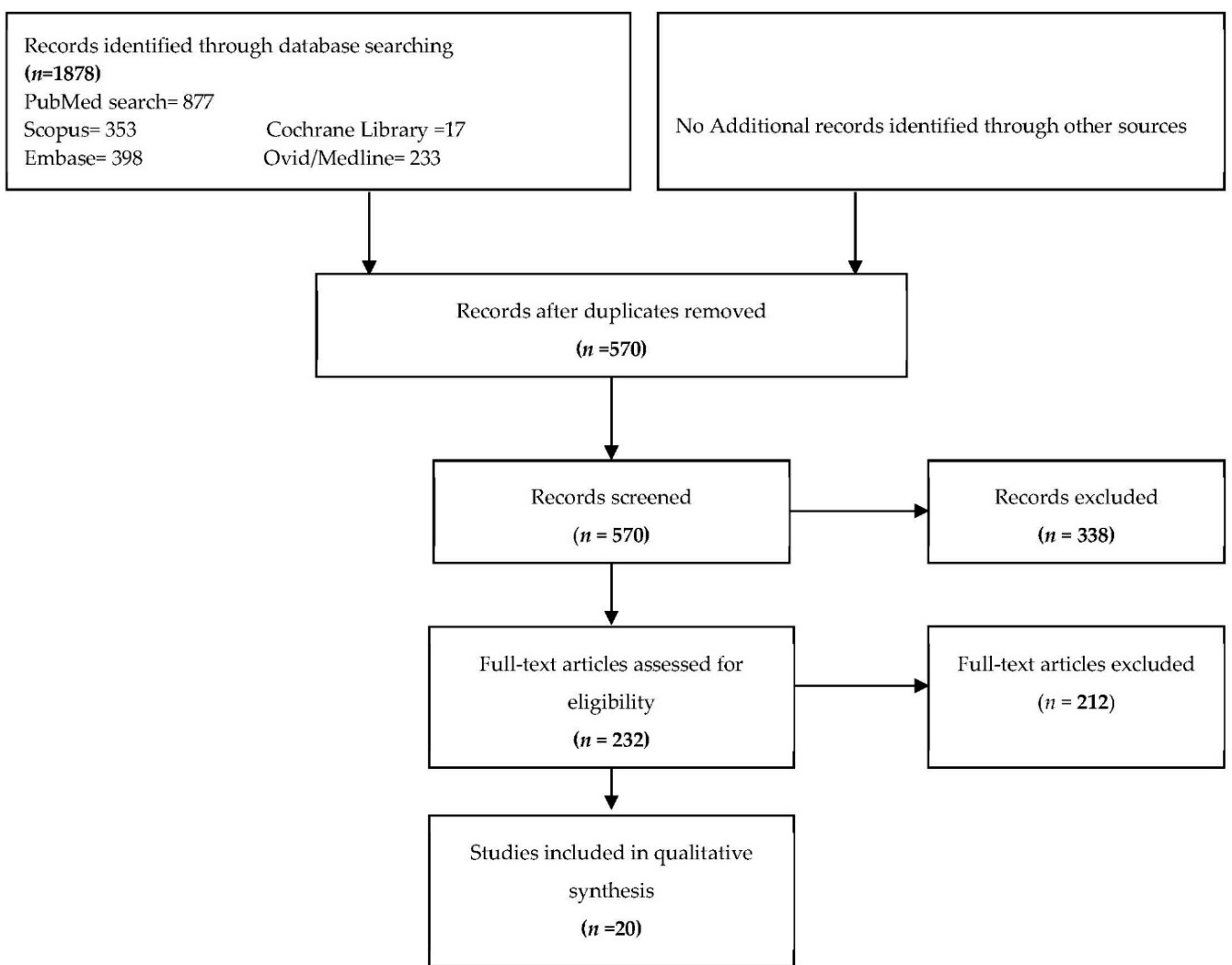

**Figure 1.** Flowchart of the screening process.

### 3.1. Quality of Included Studies

In general, the studies included in the compilation were cross-sectional or case-control type studies. Individually, the studies explored different testing methodologies and biological substrates (GCF, blood, saliva, etc.) to compare the CD14 expression. Thus, as mentioned earlier, the results were tabularized and categorized into the following four substrate subgroups: CD14 in peripheral monocytes, serum sCD14, saliva/GCF, and biopsy studies. In order to allow a more specific assessment and evaluation of the variations in technical or interexperimental design, samples, and outcomes.

Of the 20 included studies ([18–37]), 14 ([18–22,24,27,28,30,32,34–37]) were deemed of moderate quality, and 6 were categorized as low quality [23–25,29,31,33]. Reasons for classifying as low-quality involved indirectness of evidence (missing controls), extreme small sample sizes, and/or limitations or errors in design.

### 3.2. mCD14 in Peripheral Monocytes

Buduneli et al. [19], Nagasawa et al. [22], Pietruska et al. [21], Nicu et al. [20], Cheng et al. [18], and Shapira, Soskolne and Van Dyke [24] analyzed the expression of mCD14 in peripheral monocytes (Table 1). The expression of mCD14 in peripheral monocytes was measured and analyzed through in-cell ELISA and/or three-color flow cytometry [18–22]. Shapira, Soskolne and Van Dyke [24] found a decrease in mCD14 expression on monocytes in aggressive periodontitis (AP) patients compared to healthy control subjects as determined by in-cell ELISA. Nicu et al. [20] measured mCD14 expression in polymorphonuclear cells (PMNs) and monocytes in response to bacterial stimulation with *Aggregatibacter actinomycetemcomitans* and *P. gingivalis* by flow cytometry and found that cells isolated from periodontal patients show concomitant low mCD14 [20]. Buduneli et al. [19], using flow cytometry to analyze peripheral blood mononuclear cells (PBMCs) from patients and control subjects, found that the level of mCD14 in early onset periodontitis patients (7.18%, age 15–35 years) was lower than that of adult periodontitis patients (9.3%, age 38–57 years) and the control subjects (9.2%, age 22–54 years). Nagasawa et al. [22] using a similar method found that the percentage of CD14$^{bright}$ monocytes in chronic periodontitis (CP) patients was significantly lower than healthy subjects. Pietruska, Zak, Pietruski and Wysocka [21] explored the expression of mCD14 and reported that the expression of mCD14 on monocytes in generalized AP patients and control subjects were comparable [21]. Finally, Cheng, Saleh, Abuaisha Karim, Hughes and Taams [18] found no significant difference in mCD14 expression in PBMCs from patients with CP or AP patients and periodontally healthy control patients.

### 3.3. CD14 and Serum

Hayashi, Masaka and Ishikawa [36], Nicu, Laine, Morré, Van der Velden and Loos [35], Raunio, Knuuttila, Karttunen, Vainio and Tervonen [37], and Pietruska et al. [21] analyzed the expression of sCD14 in serum using ELISA (Table 2). Hayashi, Masaka and Ishikawa [36] showed that the sCD14 levels in the sera of patients with periodontitis were significantly higher than those of healthy subjects [36]. Furthermore, periodontal therapy resulted in a decrease of sera sCD14 levels in periodontitis patients [36]. Nicu, Laine, Morré, Van der Velden and Loos [35] showed that periodontitis patients have a tendency towards increased sCD14 levels in comparison to healthy controls ($p = 0.061$). However, the application of frequency distribution analysis illustrated that sCD14 levels of moderate and severe periodontitis patients had higher sCD14 values above threshold of 2.03 mg/L compared to healthy controls [35]. Moreover, patients with moderate and severe periodontitis exhibited a positive correlation between sera sCD14 levels and the severity of the periodontal destruction [35]. Finally, Raunio, Knuuttila, Karttunen, Vainio and Tervonen [37] reported that the mean concentration of sCD14 in serum was also significantly higher in subjects with periodontitis than in control subjects, and was significantly associated with the extent of advanced periodontal disease [37]. In contrast, Pietruska et al. reported similar levels of sCD14 in serum of periodontitis patients and healthy subjects.



### 3.4. CD14 in Saliva and Gingival Crevicular Fluid

Isaza-Guzman, Aristizabal-Cardona, Martinez-Pabon, Velasquez-Echeverri and Tobon-Arroyave [26], Prakasam and Srinivasan [27], Jin and Darveau [25], and Duncan, Yoshioka, Chandad and Grenier [23] analyzed CD14 in saliva and GCF (Table 3). The analysis of sCD14 was done in whole saliva samples [26,27] and gingival crevicular fluid (GCF) samples [23,25].

Isaza-Guzman, Aristizabal-Cardona, Martinez-Pabon, Velasquez-Echeverri and Tobon-Arroyave [26] obtained unstimulated whole saliva samples from patients with CP and AP, and healthy controls and measured salivary levels of sCD14 by ELISA. The levels of sCD14 were found significantly different among the three clinical groups: Significantly greater values were found in the CP and AP groups compared with healthy controls, whereas no significant difference between CP and AP patients was observed. Furthermore, a highly significant correlation between salivary sCD14 levels and the clinical measurements, such as clinical attachment loss, mean pocket depth and the percentage of periodontal pocket ≥4 mm deep and attachment loss ≥2 mm was observed. Prakasam and Srinivasan [27] analyzed sCD14 in unstimulated whole saliva by ELISA and found significantly higher sCD14 levels in CP patients than healthy controls. Scaling and root planning led to the gradual decrease in salivary sCD14 levels within 6 weeks post-therapy [27].

Jin and Darveau [25] measured the levels of sCD14 in GCF by an ELISA based capture assay in 15 Chinese adults with untreated moderate to advanced periodontitis. The results indicated that sCD14 was detected in all 15 subjects and sCD14 concentration had a significant negative correlation with pocket depth. Duncan, Yoshioka, Chandad and Grenier [23] analyzed the presence of soluble CD14 in GCF by Western immunoblotting. In this case, results showed that 3 out of 8 GCF samples from healthy periodontal sites had traces of soluble CD14, whereas 9 out of 16 samples from patients affected by moderate or advanced periodontitis contained a higher amount of soluble CD14.

### 3.5. CD14 in Biopsy/Tissues Studies

Sumedha, Kotrashetti, Nayak, Nayak and Raikar [28], Nibali, Novoa, Donos, Henderson, Blanco and Tomas [29], Li, Chen, Ng, Fung, Xu, Cheng, Tsao and Leung [30], Roberts, McCaffery and Michalek [34], Scheres, Laine, Sipos, Bosch-Tijhof, Crielaard, de Vries and Everts [31], Ren, Leung, Darveau and Jin [32], and Jin, Ren, Leung and Darveau [33] analyzed the expression of CD14 in gingival biopsies and periodontal tissues. More precisely, the expression of CD14 was explored in gingival tissue samples [28–30,34], periodontal ligament fibroblasts [31], and gingival pocket biopsies [32,33].

Sumedha et al. reported that CD14 expression in gingival tissues as assessed by immunohistochemistry was greatest in the periodontitis group that was classified as a severe grade, followed by moderate and mild grades expression and minimal in healthy gingiva [28]. Nibali et al. showed that gingival biopsies of diseased sites of periodontitis patients contained an enhanced number of CD14 compared to the control sites [29]. Furthermore, it was also found that sampled sites with less bleeding on probing exhibited higher expression of CD14 than those from 2–3 bleeding sites [29]. Li, Chen, Ng, Fung, Xu, Cheng, Tsao and Leung [30] performed immunohistochemistry, RT-PCR, and immunoblotting on gingival biopsies of patients with advanced CP and control sample biopsies. They showed that CD14 was detected in both inflamed and healthy gingival tissue. Even though the authors did not attempt to quantify CD14 expression, the research group mentions it did not observe any varied detectability of CD14 in healthy and periodontitis tissue biopsies [30]. Finally, a reverse transcription PCR assessment of gingival mononuclear cells and whole gingival biopsies from 32 adult periodontitis patients and five healthy individuals for inflammatory cytokines showed that CD14 was present in both samples [34]. Tissue section histochemistry illustrated that CD14 was detected in very low numbers in healthy biopsies and more prevalently in diseased biopsies [34]. Explaining further that the CD14-positive cells were primarily in the connective tissue and in close association with the lymphoid aggregate [34].

Scheres et al. analyzed the gene expression of CD14 in the primary periodontal ligament and gingival fibroblast cells from periodontitis patients and healthy controls [31]. Periodontal ligament fibroblasts from periodontitis patients had a higher basal and *P. gingivalis* induced mRNA expression of CD14 than healthy controls, whereas no difference in gingival fibroblasts was detected [31]. Ren et al. used immunohistochemical staining of gingival biopsies and periodontal pocket tissue from subjects with CP to explore the location of mCD14 and related proteins [32]. Outcomes illustrated that mCD14 was mainly confined to the cells around epithelium-connective tissue interface in all samples [32]. Moreover, CD14 was co-detected in 89% of the samples with lipopolysaccharide binding protein, and mCD14 peptide expression levels were significantly higher in healthy controls than those in diseased tissues [32]. These results were similar to Jin, Ren, Leung and Darveau [33], where gingival biopsies were obtained and mCD14 was also detected by immunohistochemistry. This study showed that the mCD14-positive cells were mainly confined to the gingival epithelium-connective tissue interface. However, the expression levels in periodontally healthy tissues showed greater levels of mCD14 than periodontal pocket tissues and granulation tissues [33]. mCD14 was frequently expressed in both healthy and diseased gingival tissues and was primarily confined to the epithelium-connective tissue boundary [33].

## 4. Discussion

Disease manifestations through bacterial infections can take various forms including, but not limited to, direct effects of bacterial products, alterations of the host in response to the organism, and/or the persistent response of the hosts immune system after the clearance of the organism [55]. Though the mechanisms of the inflammatory host response have been extensively studied, CD14 remains a fascinating molecule whose functions and associations in role to periodontitis remain somewhat elusive. CD14 serves as an assessor molecule for different TLRs and facilitates the recognition of different bacterial components [56,57]. Additionally, CD14$^+$ monocytes are important precursor cells from osteoclast development, which subsequently, in combination with osteoblasts, play an integral role in bone homeostasis. Hence further solidifying its role periodontal disease. Recently the role of sCD14 transducing and activating passage ways that could influence osteoclast genesis have been proposed [58].

In this systematic review, peripheral monocytes, serum, saliva, GCF, and tissue biopsies were chosen as subsets, as they are all thought to reflect different aspects of inflammatory response. Prior studies suggest that there is an increase in the percentage of monocytes in the peripheral blood of patients with chronic periodontitis [59]. Saliva has been actively scrutinized for its potential in primary diagnostics, exposing possible markers for early diagnosis of periodontal disease [60]. GCF is a filtrate of blood and an exudate of the inflamed periodontal tissue [61]. It has been used as a promising site-specific intermediate that exposes how the local host responses react to microbial challenges such as it can be found in periodontitis [62]. Thus, it was the goal of this systematic review to assess and examine local sCD14 and mCD14 levels in different tissues, physiological fluids, and periodontal compartments.

Membrane-bound CD14 has been thought to only be expressed on circulating monocytes and tissue macrophages [9]. It is one of the factors, which facilitates the recognition of LPS by TLR-4 of host cells [10]. The expression of mCD14 in peripheral monocytes was analyzed, and results indicate that in four reports the expression of mCD14 on monocytes is decreased in periodontitis [19,20,22,24], while two additional reports note no difference [18,21].

In contrast to mCD14 expression in peripheral blood monocytes, the majority of studies on sCD14 levels in various biological fluids show a positive association between sCD14 levels and periodontal disease. This might be interpreted as an increase in production of sCD14 in cases with moderate-to-severe and generalized periodontal breakdown. The local levels of sCD14 measured in saliva and GCF are also affected by periodontitis. In two studies enhanced levels of sCD14 in periodontitis patients compared to healthy controls

were found [26,27]. Furthermore, scaling and root-planning led to the decrease of salivary sCD14 levels, which underlies a positive association with periodontal disease [27]. The data on sCD14 levels in GCF are less conclusive [23,25].These inconsistencies could be explained by differences in small sample size, variations in individual profiles, cite specific difference in GCF composition and generally complicated procedure of GCF collection [63].

Interestingly, one study also hypothesized and explored the concept of in vivo cleavage of CD14 during periodontitis [23]. Particularly, immunoreactive CD14 fragments were present in certain GCF samples of advanced (pocket depth ≥7 mm) periodontitis patients. Further, susceptibility to degradation by *P.gingivalis* outer membrane vesicles was demonstrated, indicating that the CD14 receptor was highly sensitive to the proteinases associated with *P. gingivalis*. Thus, one could hypothesize that cleaving of mCD14 from monocytes might result in decreased phagocytosis and pathogen elimination, subsequently allowing easier reproduction of bacteria through a "weakening" of CD14 efficacy. Recent studies could support this ideas, as one study indicated that *P. gingivalis* gingipains can selectively reduce the responsiveness of macrophages to *P. gingivalis* infection [64]. Another recent study reported that the amount of mCD14 on peripheral blood monocytes treated with *P. gingivalis* proteinases was reduced [65]. In turn, increase of sCD14 or its fragments due to degradation of mCD14 by *P. gingivalis* might sensitize connective tissue cells to LPS and lipoproteins [14,15]. This will result in promoting inflammatory response and tissue destruction, and subsequently an enrichment with host derived protein as nutrition for bacteria.

Studies on the tissue biopsies report different results regarding the expression of CD14, yet suggest in most cases an enhanced CD14 expression in periodontitis. It is important here to note the differences in methodology applied, the type of biopsy and tissue tested as well as the type of CD14 molecule explored. Moreover, the expression of CD14 might be tissue specific.

Noteworthy, none of these five biopsy studies discriminated between sCD14 and mCD14. Although some studies mentioned that they use anti-mCD14 antibodies (e.g., [32,33]), it is difficult to imagine that these antibodies might differentiate between mCD14 and sCD14. Similarly, a distinguishing between these two CD14 forms is hardly imaginable in studies using PCR technique [31,34].

Taken together, these are all cross-sectional studies and limitations are inherent with their nature. The frequent small sample sizes, differences or flaws in sample collection, or lack of inclusion of healthy control subjects can limit studies. Additionally, contributing factors, processing errors, or limitations due to unavailable data and the mere adaptions in definitions each research team choose to qualify as healthy or diseased could influence subsequent outcomes. For example, in the saliva/GCF studies the method and timing of sample selection comes with great outcome variability, it has been shown that accurate analysis is difficult [66]. GCF can also contain molecular products from various sources including, but not limited to, epithelium cells, inflammatory cells, and bacterial products from the crevice and surrounding tissue [25] that could cause discrepancies. Confirmatory studies with increased sample sizes and longitudinal evaluations with lengthier follow ups would need to be evaluated to prove reliability. Furthermore, correlation doesn't always apply cause and effect, thus randomized clinical studies are needed to clarify these points.

## 5. Conclusions

This review warrants that additional studies with greater sample populations and methodically standardized procedures are needed to further elucidate the molecular mechanisms.

Nonetheless, this review does highlight a few important concepts. One, that CD14 is frequently detected in healthy and diseased tissues, but CD14+ cells are predominantly confined in the gingival epithelial connective tissue interface and in lower quantities in the diseased state. Possible downregulation of mCD14 expression could be due to shedding regulated by *P. gingivalis* gingipains and cleaving of mCD14 from peripheral monocytes, consequently resulting in a slower responsiveness of macrophages and monocytes as well

as reduced phagocytosis and pathogen elimination of *P. gingivalis*. This decline in CD14 efficacy could be advantageous for the proliferation of bacteria, consequently leading to an increase in inflammation. Moreover, the findings that sCD14 levels are increased in serum, GCF, and tissue biopsies could show that it serves as a sensitizing molecule for connective tissue, which further promotes the inflammatory response, osteoclast formation, and destruction of bone and soft tissue. Increased levels of sCD14 in serum could also be due to the fact that it has been postulated that the mechanism of bacterial lytic activity is withstood in blood serum. Lastly, measuring pre- and post-treatment sCD14 concentration could perhaps make sCD14 a useful marker for monitoring development of periodontal treatment.

Considering different experimental methods, cell types, media, and reagents, this review details the alterations in CD14 in the periodontal pathogenesis, however, also reveals the contradictions and controversies that still remain. These findings further reflect the dynamic nature of the local host inflammatory response to microbial challenge and the awareness that there still are gaps in our comprehension.

**Author Contributions:** Conceptualization, V.H., A.B. and O.A.; methodology, V.H. and O.A.; validation, V.H., A.B., C.B. and O.A.; formal analysis, V.H.; investigation, V.H.; resources, O.A.; data curation, V.H.; writing—original draft preparation, V.H.; writing—review and editing, A.B., C.B., and O.A.; visualization, V.H.; supervision, O.A.; project administration, O.A.; funding acquisition, O.A. All authors have read and agreed to the published version of the manuscript.

**Funding:** Open Access Funding by the Austrian Science Fund (FWF), grant number P29440.

**Institutional Review Board Statement:** Not applicable.

**Informed Consent Statement:** Not applicable.

**Data Availability Statement:** Data sharing not applicable.

**Conflicts of Interest:** The authors declare no conflict of interest.

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
