# Peer review of "The Alterations in CD14 Expression in Periodontitis: A Systematic Review"

_applsci, doi:10.3390/app11052444_

Round 1
Reviewer 1 Report
Authors have clearly and succinctly formulated systematic review. They hypothesis, and selection of cross-sectional or case control study articles for systematic review is adequate. Definition of inclusion and exclusion criteria are acceptable. Critical and systematic review of the selected article are nicely presented. The conclusion is influenced mainly by the lack of quality of articles and hence further studies are needed to clarify the level of mCD14 vs sCD14 expression linking periodontal pathogenesis.
Minor comment:
1) CD14+ monocytes are major precursor cells for osteoclast development. Ratio of osteoclast vs osteoblast plays a crucial role in periodontal disease. I think it would be interesting to add brief discussion on this aspect and link why mCD14 and sCD14 are important in periodontal disease.
Author Response
Thank you very much for the evaluation of our work and positive feedback.
COMMENT
1) CD14+ monocytes are major precursor cells for osteoclast development. Ratio of osteoclast vs osteoblast plays a crucial role in periodontal disease. I think it would be interesting to add brief discussion on this aspect and link why mCD14 and sCD14 are important in periodontal disease.
RESPONSE
Thank you for raising this point; we discuss it in the revised version (see lines 309-313).
Reviewer 2 Report
This is a by-the -book made systematic review showing the scarcity of research on the role CD14 in the pathogenesis and diagnosis of periodontal disease. Based on the 20 original papers meeting the criteria, there seems to be a trend that the CD14 molecule can be detected in various body samples and that it reflects the disease. I only suggest condensing the Discussion and Conclusion because the results indeed do not entitle any final conclusion. Further, the list of references is not in uniform style.
Author Response
COMMENT:
This is a by-the -book made systematic review showing the scarcity of research on the role CD14 in the pathogenesis and diagnosis of periodontal disease. Based on the 20 original papers meeting the criteria, there seems to be a trend that the CD14 molecule can be detected in various body samples and that it reflects the disease. I only suggest condensing the Discussion and Conclusion because the results indeed do not entitle any final conclusion. Further, the list of references is not in uniform style.
RESPONSE:
Thank you very much for the evaluation of our work and positive feedback. In the revised version, we have shortened the discussion by about 20 %. The reference style was also improved.